# Gut Microbiome in Patients after Heart Transplantation—Current State of Knowledge

**DOI:** 10.3390/biomedicines11061588

**Published:** 2023-05-30

**Authors:** Katarzyna Olek, Agnieszka Anna Kuczaj, Szymon Warwas, Tomasz Hrapkowicz, Piotr Przybyłowski, Marta Tanasiewicz

**Affiliations:** 1Department of Dental Propedeutics, Faculty of Medical Sciences in Zabrze, Medical University of Silesia, 41-800 Zabrze, Poland; katarzynajuliaolek@gmail.com; 2Department of Cardiac Surgery, Transplantology, Vascular and Endovascular Surgery, Faculty of Medical Sciences in Zabrze, Medical University of Silesia in Katowice, M.C. Sklodowskiej 9, 41-800 Zabrze, Poland; 3Students’ Scientific Association Affiliated with the Department of Cardiac, Vascular and Endovascular Surgery and Transplantology, Faculty of Medical Sciences in Zabrze, Medical University of Silesia, 41-808 Zabrze, Poland; szymonwarwas@gmail.com; 4Department of Conservative Dentistry and Endodontics, Faculty of Medical Sciences in Zabrze, Medical University of Silesia, 41-800 Zabrze, Poland; martatanasiewicz@sum.edu.pl

**Keywords:** dysbiosis, gut microbiota, orthotopic heart transplantation

## Abstract

The human gut microbiota include over 10 trillion microorganisms, such as bacteria, fungi, viruses, archaea, and protozoa. Many reports indicate the strong correlation between dysbiosis and the severity of cardiovascular diseases. Microbiota seem to interact with the host’s alloimmunity and may have an immunomodulatory role in graft rejection processes. In our study, we present the current state of the knowledge of microbiota in heart transplant recipients. We present up-to-date microbiota diagnostic methods, interactions between microbiota and immunosuppressive drugs, the immunomodulatory effects of dysbiosis, and the available strategies (experimental and clinical strategies) to modulate host microbiota.

## 1. Introduction

Cardiovascular diseases are still the leading cause of reduced quality of life, morbidity, and mortality in developed societies [1,2]. End-stage heart failure (HF) is a severe, final stage of many cardiovascular diseases. Epidemiological data show that the incidence of heart failure varies between 1% and 2% and increases to 10% in people over 70 years of age [3,4,5]. The prognosis after the diagnosis of heart failure is poor: five-year survival is less than 50%. Although significant progress has been made in the treatment of HF, there are still two vital treatment options for this condition: left ventricular assist devices (LVAD) and heart transplantation (HT) [6,7,8]. There is much information in the literature on the importance of the gut microbiome in cardiovascular diseases, including in patients treated with LVAD and after HT [9,10,11].

The human gut microflora includes over 10 trillion microorganisms, such as bacteria, fungi, viruses, archaea, and protozoa. The microflora in a healthy person consists mainly of the following groups of bacteria: Bacteroides, Actinobacteria, Proteobacteria, Firmicutes, Verrucomicrobiota, Cyanobacteria, Fusobacteria, Spirochaetes, and Vadin BE97, with Bacteroides and Firmicutes accounting for 90% of the total bacterial population. The phylum Firmicutes consists of 95% Clostridium, while the majority of Bacteroides are Prevotella and Bacteroides. This composition of microorganisms is constantly modified [12]. Reduced microbial diversity comprising the depletion of potentially beneficial saprophytic microbiota (Bacteroides and Firmicutes) and the overgrowth of pathogenic bacteria is called dysbiosis [13].

Evidence from several previous studies confirm the importance of the gut microflora in the transplantation of solid organs (kidneys/livers). Strong evidence indicates that the gut microflora has an impact on systemic inflammation, immune response, and infectious complications after transplantation. The disruption of physiological microflora may adversely affect the transplant results and lead to infection, fibrosis, rejection of the graft, and changes in drug metabolism. A proper diet, probiotics, and prebiotics can influence intestinal dysbiosis [14,15,16,17].

In the current study, we try to focus on the role of gut microbiota in patients with end-stage heart failure treated with heart transplantation. We analyze and summarize different aspects of this issue. We present the current diagnostic methods, natural history of microbiome changes after heart transplantation, impact of gut microbiota on inflammatory and rejection processes, and available therapeutic methods in this area.

## 2. Current Diagnostic Methods of Gut Microbiota

In the past, the gut microbiome was analyzed using isolation and culture. Due to the difficulties in culturing anaerobic bacteria, the accuracy of the analysis was questionable.

There are many methods of sampling for the analysis of intestinal microflora. We will briefly discuss each of them.

1.Samples from feces

This is a non-invasive, natural, inexpensive test that can be repeated many times. However, it should be noted that there are significant differences between the microflora found in feces and that found in the intestinal mucosa. Recent studies have shown that fecal microflora, compared to that associated with mucous membranes, is composed of two distinct microbial niches [18,19]. In addition, the fecal microflora is not evenly distributed in the feces and has its own biostructure. Different bacteria are found in the small intestine, and others in the final section of the digestive tract. The need for the proper storage of fecal samples should also be considered [20].

2.Samples from endoscopy:
BiopsyFew studies have been conducted on endoscopy samples. Much has been said about the disadvantages of this method. Among other things, it is invasive, expensive, time consuming, and unpleasant for patients.Before the endoscopy, the patient should be adequately prepared for the examination. Laxatives, such as polyethylene glycol, should be given, which significantly impacts the gut microflora. In addition, using endoscopy, we are not able to reach the final section of the small intestine, and the material is only collected from a small area of the intestine, which can give selective results. Moreover, the collected material may be contaminated with the fluid in the sampling device. To minimize the risk of sample contamination, a unique biopsy device (Brisbane Aseptic Biopsy Device (BABD)) has been developed, which consists of sterile forceps covered with a sheath and sealed with a plug at the ends. The advantage is a controlled sampling site and the ability to obtain an accurate description of the tissue-associated microflora [21,22].Luminal brushingThe protected specimen brush (PSB) is a disposable, sterile brush that is housed in a special cover with a cap, which is sealed when inserted through the colonoscopic canal. Compared to a biopsy, brushing the intestinal mucosa reduces the risk of bleeding or infection and is less invasive. Additionally, luminal brushing contains a relatively high ratio of bacterial DNA to host DNA. The advantage is, as in the case of a biopsy, a controlled sampling site and the possibility of obtaining an accurate description of the microflora associated with the tissue. Samples are also taken during the endoscopic examination; therefore, this method has the same disadvantages as a biopsy—the intestine must be adequately prepared, which results in a change in the diversity of the microflora. It is also an invasive, time-consuming, and expensive method [23,24].Laser capture microdissection (LCM)The source of the sample is also a biopsy and therefore suffers from the same disadvantages as the two previously mentioned methods. Then, the collected material is properly and precisely prepared—this limits the use of the method on a large scale. Then, with the help of a laser, the microflora is carefully analyzed. The advantage is an accurate representation of the interaction between the host and the microbiome, as well as a controlled sampling site [23,25].

3.Samples from aspirated intestinal fluid

This method involves suctioning out the intestinal fluid. At present, samples of aspirated intestinal fluid are collected by endoscopic aspiration. This method has the same disadvantages that have been mentioned with endoscopy—we are not able to reach the final segment of the small intestine, and the material is only collected from a small area of the intestine, which can give selective results. In addition, the collected material may be contaminated with fluid in the sampling device. It is also a time-consuming procedure, causing patient discomfort. The advantage is the ability to obtain a precise description of the luminal microflora and a controlled sampling site [26].

4.Samples from surgery

Unlike other methods, this method enables the collection of material from the final section of the small intestine. The material during the procedure is collected by needle aspiration or biopsy. It is said that samples taken using this method best reflect the composition of the microflora and are not contaminated. We also control the exact place where the sample was taken. On the other hand, the proper preparation of the patient for the procedure, in the form of administering, for example, laxatives or antibiotics, may significantly disturb the composition of the gut microflora. It is a very invasive method [26].

5.Ingestible sampling devices

This is a non-invasive method, consisting of the patient swallowing a special capsule, which aspirates the food in the right place in the intestine. The aspirated fluid can be collected after the capsule is emptied from the intestine. Thanks to the lack of the necessity to prepare the intestine, it is a more accurate and precise method—it does not change the composition of the gut microflora. Thanks to this method, we can obtain an accurate description of the luminal microflora. There is also less risk of contamination of the collected sample. It is a relatively expensive and technically complicated method [27].

6.In vivo model (patients who underwent ileostomy)

This method is dedicated to patients who have had an ileostomy. This procedure significantly changes the anatomical structure of the intestine, which can have a significant impact on the composition of the gut microflora. It is a relatively inexpensive, non-invasive method. We can perform reproducible sampling (sufficient biomass for analysis) and minimize the risk of sample contamination [28].

7.Biology-related instruments

The method used to identify microorganisms is the FISH method (fluorescence in situ hybridization), which, with the usage of a fluorescence microscope, allows us to accurately identify the spatial organization of the microbiome. Improved histological preparations allow, in addition to identifying microbial diversity, the ability to determine the host–microbiota correlation. Due to ethical problems, difficulties in sampling (the probe used to collect the material must be designed in advance), and high individual variability in the microbial composition, the application of this method in situ is limited [29].

At the moment, there is no single, ideal method for collecting samples of the intestinal microbiome. Research on this is still ongoing.

The current gut microbiota diagnostic methods and their advantages and disadvantages are summarized in Table 1.

## 3. Dysbiosis Due to Different Immunosuppressive Drugs

Immunosuppressive therapy is an inseparable element of treatment after OHT to prevent graft rejection, and in the vast majority of cases, it is a lifetime treatment. These actions could potentially influence patients’ microbiota diversity. Immunosuppressive drugs may affect the composition of the gut microbiome by selectively inhibiting growth and promoting bacterial strains [30]. An in vitro study on over 1000 non-antibiotic drugs showed that almost ¼ of them (including immunosuppressive drugs) constrain bacterial growth [31]. Immunosuppression after OHT consists of calcineurin inhibitors such as tacrolimus or cyclosporine, antimetabolites such as mycophenolate mofetil/mycophenolic acid (MPA), and mTOR inhibitors such as everolimus and glucocorticosteroids. Each of them has a different impact on the gut microbiome. Tacrolimus was found to alter the diversity of gut microbiota in different mechanisms. It leads to dysbiosis by lowering ileal Regenerating Islet-Derived Protein 3 Beta levels and increasing gut permeability [32]. It has been proven that tacrolimus, when given orally in mouse populations, causes a significant increase of Firmicutes, Bacteroidetes, Allobaculum, and Lactobacillus; however, the abundance of Rikenella, Clostridium, Oscillospira, and Ruminococcaceae seems to decrease substantially [15,33]. On the other hand, Bhat et al. showed that intraperitoneal injections of tacrolimus did not alter the Firmicutes to Bacteroidetes ratio; however, they observed Actinomycetales, Rothia, Staphylococcus, Roseburia, Mollicutes, Oscillospira, and Micrococcaceae to be less plentiful [34]. The interesting fact is that Faecalibacterium prausnitzii, a species in the class of Clostridia and a common type of gut bacteria, is capable of converting tacrolimus into 15-fold less effective metabolites [35]. It was also shown that its abundance is significantly higher in kidney transplant recipients receiving tacrolimus [36]. It was also discovered that gut microbiota change with different schemes of tacrolimus dosage after OHT. Jennings et al. found that a higher posology of tacrolimus was correlated with a more diverse composition of gut microbiota [37]. Except for tacrolimus, cyclosporine A is a common immunosuppressive drug used in heart transplant recipients. In the rat model, cyclosporine decreased the proportion of Enterobacteriaceae and Clostridium clusters I and XIV with the increase in Faecalibacterium prausnitzii [38]. On the other hand, the human model with the implementation of encapsulated cyclosporine showed no alteration of gut microbiota diversity [39]. Moreover, several studies showed that introducing MPA favors the development of dysbiosis. Flannigan et al. found that exposing mice to MPA leads to the loss of the diverse composition of microbiota with a significant increase in Proteobacteria, a decrease in Akkermansia, Parabacteroides, and Clostridium, and also the appearance of multiple lipopolysaccharide biosynthesis genes. It resulted in the rapid weight loss of the mice and general intestine inflammation [40]. In contrast, another study in mice revealed the expansion of *Clostridia* and *Bacteroides* spp. [41]. The influence of everolimus, the mTOR inhibitor, on the gut microbiota seems to be insignificant [17]. In the study by Zaza et al., they compared microbiota compositions in patients after renal transplantation, finding no significant differences between the tacrolimus group and the everolimus group [42]. Glucocorticosteroids, another crucial group of drugs in preventing graft rejection, have a diversified impact on gut microbiota. In the rodent model in which they received liver transplantation, the implementation of prednisolone was found to alter the ratio of Bacteroidetes and Firmicutes in favor of the latter [17]. Dexamethasone, the other commonly used steroid, turned out to delay the recovery of gut microbiota compositions previously amended by the usage of antibiotics [43]. Apart from disruptions in gut microbiota composition, immunosuppressants may also have a positive impact on the homeostasis of the intestines, as they seem to revert the dysbiosis associated with graft rejection [38]. Significant concerns should be raised towards the influence of combined immunosuppressive therapy on gut bacterial populations since the treatment after organ transplantation is almost never a monodrug therapy. Tourret et al. showed that three-drug therapy consisting of tacrolimus, prednisolone, and MPA significantly decreased Clostridium sensu stricto genus abundance [17].

Due to the fact that the results of the following studies often reveal ambiguous conclusions, there is a need for further research in order to ascertain what mechanisms underline the influence of immunosuppressive therapy on the composition of gut microbiota. 

## 4. Changes in Gut Microbiota over the Course of Time

As previously mentioned, one of the main reasons for dysbiosis after organ transplantation is the implementation of immunosuppressive therapy. Immunosuppressive drug treatment changes over the course of time after transplantation; thus, gut microbiota should be altered in a different manner with time. Little is known about how gut microbiota change gradually after OHT because most studies focus on changes in liver and kidney recipients. 

Recently, the concept of the heart–gut axis has played an increasingly significant role in the understanding of HF pathogenesis. Patients with HF suffer from reduced cardiac output, which translates into peripheral perfusion disturbances resulting in gut function impairment, increased intestinal permeability, and tissue congestion [44]. An increased bowel wall thickness of the colon and terminal ileum can be also found in patients with HF, which might suggest the development of bowel edema in them. Apart from that, HF patients have an increased density of bacterial biofilm and higher concentrations of adherent bacteria with augmented intestinal permeability, which may lead to the translocation of bacteria from the gut to the bloodstream, resulting in chronic inflammation [44,45].

There were several studies in which a correlation between gut microbiota dysbiosis and the severity of heart failure (HF) was assessed. Nagamoto and Tang underline that patients with heart failure may have an altered gut microbiota richness due to intestinal hypoperfusion, especially in the phase of decompensation [46]. In addition, Kummen et al. showed a depletion of butyrate-producing gut (BPG) bacteria (mainly the Lachnospiraceae family), which could play a role in the development of inflammatory changes [47]. Intestinal epithelial cells derive energy from butyrate metabolism; thus, its deficiencies negatively influence the proper functioning of the intestinal epithelial barrier and the progression of intestinal inflammation [48]. Apart from this, in the study of Luedde et al., they found a significant reduction of bacteria belonging to the families of Ruminococcaceae, Coriobacteriaceae, and Erysipelotrichaceae [49]. On the other hand, an Italian study on 60 patients with HF showed an expansion of Candida and pathogenic gut bacteria, such as Campylobacter, Yersinia enterocolitica Shigella, and Salmonella [50]. There were also observations in a mouse model that strongly indicated that dysbiosis is associated with more severe HF [51]. 

Yuzefpolskaya et al. performed a study of over 450 patients with heart failure, those with left ventricle assist devices, and heart transplant recipients, revealing a decrease in microbial diversity within six months after OHT. The other finding was that the abundance of gut populations decreases with the progression of the heart failure class, mainly affecting the family of Lachnospiraceae and Ruminococcaceae. Patients in higher NYHA classes also had significantly expressed markers of inflammation such as C-reactive protein, tumor necrosis factor-alpha, or interleukin-6 with a substantial level of bacterial lipopolysaccharide (LPS), soluble CD14, and an elevated level of isoprostane as a biomarker of oxidative stress. However, after the implementation of mechanical circulatory support or OHT, the levels of these markers declined, but the level of endotoxemia was not diminished [52]. Another study also showed that systemic inflammation in patients with decompensated heart failure is linked with higher LPS levels in their blood. Nevertheless, with the patients’ improvement, the LPS levels gradually diminished [53]. 

As previously mentioned, one of the main reasons for dysbiosis after organ transplantation is the implementation of immunosuppressive therapy. Immunosuppressive drug treatment changes over the course of time after transplantation; thus, the gut microbiota should be altered in a different manner with time. Little is known about how gut microbiota change gradually after OHT because most studies focus on changes in liver and kidney recipients. Swarte et al. found that dysbiosis can occur up to six years after renal transplantation [54]. On the other hand, Wu et al. found that gut bacteria populations, except for *Enterococcus* spp., have a tendency to be restored after 13–24 months post liver transplantation [55]. Another study revealed that in the early stage after liver transplantation, the gut bacteria diversity decreases up to 3 weeks after surgery and then is gradually restored within 2 months of observation [56]. In kidney transplant recipients, the population was observed up to 6 months after surgery and the main changes in gut microbiota were seen during the first month post transplantation [57]. In general, solid transplant recipients’ gut microbiota are altered the most in the early stages after surgery, with gradual stabilization in the late post-transplant period.

## 5. Consequences of Gut Microbiota Dysbiosis in Heart Transplant Recipients and Infectious Complications

To prevent or treat an infection after OHT, patients undergo significant courses of antimicrobial therapy, which might lead to the colonization of multidrug-resistant pathogens such as Clostridium difficile, resulting in dysbiosis. This may bring a paradoxical effect where the usage of antibiotics results in an increased risk of infection. What was also mentioned before the implementation of immunosuppressive agents may impact the gut microbiota, resulting in an increase in pathogenic strains. In the study performed by Bruminhent et al., Clostridium difficile infection (CDI) was identified as an independent risk factor for mortality in the heart transplant population and was observed more frequently in patients undergoing re-transplantation [58]. It was established that CDI occurs most frequently within the first month after OHT [59]. In the liver transplant population, the colonization of multidrug-resistant strains was associated with a diminution of gut microbiota diversity. Apart from that, the authors suggested that the vast presence of multidrug-resistant bacteria in the gut may be a marker of persistent dysbiosis [60]. 

Tourret et al. showed that combined immunosuppressive therapy consisting of tacrolimus, MPA, and prednisolone may lead to the high prevalence of urinary tract infections by increasing the amount of uropathogenic Escherichia coli strain 536 [17]. This combination of drugs is commonly used in heart transplant recipients, which should bring more attention to managing this group of patients. 

Except for bacteria, the significant issue in transplant recipients remains viral infections. Lee et al. showed that the abundance of BPG bacteria significantly decreases the risk for the development of upper respiratory tract rhinoviral and coronaviral infections up to two years post transplantation. Cytomegalovirus viremia was also decreased in the presence of high quantities of BPG bacteria one year after kidney transplantation [61]. Similar results can be observed in the study performed by Haak et al. in the group of allogeneic hematopoietic stem cell transplant recipients [62]. Patients with an increased amount of BPG bacteria in the fecal microbiota presented augmented resilience against viral infections of the lower respiratory tract.

To conclude, the significance of a balanced gut microbiota both before and after transplantation in order to avoid potentially fatal infections and slow the growth of antibiotic resistance among bacteria remains a crucial therapeutic challenge.

## 6. Pro and Anti-Inflammatory Effects of Gut Microbiota

Much has been said in the literature regarding the contribution of the gut microbiome to the pathophysiology of metabolic and cardiovascular diseases, including organ transplant patients. Intestinal dysbiosis, as congestion of the intestinal wall and hypoperfusion, can be the cause of the pathogenesis and progression of many pathologies. It promotes the translocation of bacteria and their by-products into the circulation, secondary to intestinal barrier dysfunction.

Several microbe–host pathways have been implicated that link the gut microbiome to heart failure. There is talk of the role of three routes: the overgrowth of bacteria producing lipopolysaccharides (LPS), the decrease in SCFA (short chain fatty acids)-producing bacteria, and the microbial-dependent production of pro-inflammatory/fibrotic uremic toxins such as trimethylamine-N-oxide (TMAO) [10,14].

LPS is a virulence factor that is present in the outer membrane of Gram-negative bacteria. It has been proven that the amount of lipopolysaccharide in the digestive tract >200–300 mg could cause death. A preventive factor is an appropriate intestinal barrier that prevents systemic dissemination. In heart failure, the disruption of the intestinal barrier leads to the entry of lipopolysaccharide into the bloodstream. LPS then initiates systemic inflammation by activating cytokines such as IL-6 and TNF-α. This is mediated by NF-kB and leads to apoptosis and myocardial fibrosis [63,64].Short-chain fatty acids such as butyrate, acetate, and propionate are produced in the distal large intestine. The key role of SFCA in the modulation of host immune cells was confirmed. Reduced SCFA production causes a change in the ratio of Firmicutes to Bacteroides. Such changes have been described in hypertensive patients. On the other hand, the administration of SCFA has a positive effect on the ratio of Firmicutes to Bacteroides bacteria, thus leading to a decrease in blood pressure. SCFA has also been reported to improve heart function in patients after myocardial infarction. The increased level of propionate plays a key role here. The key role of SCFA in the modulation of immune cells is also confirmed. Through histone hyperacetylation and signal transduction, regulatory T cells (Tregs) and CD4 T cells are differentiated. Inflammation is then reduced by modulating the expression of IL-10 and transforming growth factor-β. On the other hand, low levels of butyrate can initiate pro-inflammatory reactions [47,65,66,67].Trimethylamine N-oxide is formed by the bacterial conversion of choline, phosphatidylcholine, carnitine, and betaine to trimethylamine (TMA). It is absorbed in the intestine and oxidized by endogenous enzymes in the liver to TMAO. Little is known about which bacterial strains promote TMA production. Elevated levels of TMAO are associated with an increased risk of cardiovascular events such as myocardial infarction, stroke, and even death. Drugs used in heart failure do not affect the concentration of TMAO. In a mouse model fed a diet with high choline content, cardiac fibrosis and adverse remodeling of the heart ventricles occurred. The exact mechanisms of TMAO’s correlation with cardiovascular disease are not well understood [64,68,69].

Recent studies indicate that gut microbiota play a significant role in the success of the transplant, thanks to the recipient’s innate and adaptive immune systems. The microbiota can alter the host’s immune response through various signaling pathways such as TLR9 and Myd88. T lymphocytes (including regulatory T lymphocytes (Tregs)) and natural killer cells (NK) play a role in the event of organ transplant dysfunction or rejection [65]. 

## 7. Interactions between Gut Microbiota and Graft Rejection

Gut microbiota, despite their distal localization to the transplanted heart, may have an impact on graft rejection or failure. In general, the most important mechanism in which microbiota interact with transplanted organs is the modulation of alloreactivity and participation in immunosuppressive drug metabolism. Some germs influence graft tolerance, some promote graft alloimmunity, and some are neutral in the aspect of rejection [70].

The heart, in opposition to the lungs and gut, is sterile and non-colonized by the bacteria organ. The interaction between the graft antigens and host microbiota occurs indirectly via the recipient’s immune system and bacteria-derived substances. Studies have shown that some microbiota species promote T-cell-dependent IgA responses and dendritic cell licensing [71]. Another described mechanism is the induction of T regulatory cells [70].

Microbiota change after transplantation in the course of time and this depends on the use of prophylactic and therapeutic antibiotics. The antibiotics prescribed in the course of post-transplant therapy disturb and reduce the diversity of the recipient’s microorganisms and lead to dysbiosis. Dysbiosis, in turn, leads to a weakened antiviral response of the host [72], which, in the case of immunosuppressed individuals, may have deleterious effects. Interactions between some viral infections and heart graft rejection are widely described elsewhere [73].

In renal transplant recipients, significant late rejections were associated with a decrease in Anaerotruncus, Coprobacillus, Coprococcus, and Peptosreptococcaceae in rectal samples [57].

In small bowel transplantation, the dysbiosis associated with an increase in Enterobacteriaceae, especially the species Escherichia coli and Klebsiella pneumoniae, and a reduction in phylum Firmicutes and Lactobacillaes [74] is connected with rejection processes.

Available are murine models of skin and heart transplant in germ-free mice and models with pretreatment of donor and recipient with broad-spectrum antibiotics. It was observed that in germ-free mice, minor antigen-mismatched skin grafts and major antigen-mismatched (MHC II) heart grafts have prolonged survival and delayed rejection processes [75]. The pretreatment of the donor and recipient with antibiotics attenuated the alloreactive T cell response by antigen-presenting cells [75]. This observation suggests a possible role of some host microbiota in the alloreactivity mechanism.

In the experimental mice heart transplant model, Listeria monocytogenes infection led to the loss of previously established graft tolerance and promoted acute graft rejection. Listeria activated a proinflammatory state by the activation of alloreactive T cells. Major cytokines connected with the loss of tolerance were IL-6 and IFN-β [76]. Similarly, Staphylococcus aureus prevented the induction of graft tolerance by IL-6 upregulation [77], and Listeria monocytogenes enhanced alloreactivity independently from the cross-reactivity of memory T-cells [78]. These authors conclude that exposure to bacteria can antagonize tolerogenic mechanisms and enhance antigen-specific responses.

A murine model of vascular rejection showed that antibiotic treatment leading to the disruption of the original microbiome led to exaggerated vascular rejection; however, it did not influence the development of donor-specific antibodies [79].

Gut microbiota can alter immunosuppressive drug metabolism and, in some cases, lead to graft rejection due to drug underdosing. The abundance of Faecalibacterium prausnitzii is connected with a need for high tacrolimus dosing in order to achieve a tacrolimus therapeutic window. The Faecalibacterium prausnitzii bacterium is an indirect indicator of a healthy colon mucosa. Diarrhea and antibiotic administration may lead to elevated tacrolimus levels in the mechanism of dysbiosis [36].

## 8. Active Immunomodulation Strategies with the Application of Microbiota

Microbiota can be a predictive marker for graft outcome. Intervention in gut microbiota can be adjunctive intervention influencing graft and patient survival. Currently, multiple microbiota-targeted therapies in order to influence graft survival are being studied [70]. Animal models showed that microbiota transfer from the donor influenced graft survival. In this study, the graft tolerance was connected with the Alistipes genus [80].

Indirect interventions influencing microbiota are the prevention of obesity, hyperlipidemia, and regular physical activity [70]. A high-fat diet alters the composition of splenic antigen-presenting cells into a more proinflammatory phenotype [75], and a high-salt diet accelerates graft rejection by the downregulation of regulatory T cells [81].

The topic of so-called probiotics is widely discussed. Multiple market products are available, but strong evidence is still lacking. There are experimental studies showing that probiotics have a diverse impact on microbiota depending on the pretreatment host microbiome. The interpersonal variability in reaction to the therapy is high [82].

In terms of diagnostic aspects, the gut mucosal microbiome only partially correlates with stool probes, and the intervention effect is difficult to assess.

In the murine model of heart transplantation, Bifidobacterium pseudolongum microbiota transfer was associated with better survival, less histologically proven rejection, and fibrosis [83]. The anti-inflammatory mechanism was explained by the stimulation of dendritic cells and macrophages to produce anti-inflammatory cytokine IL-10 and chemokine CCL19. Lesser amounts of TNF-alpha and IL-6 were observed in mice colonized with this germ. Furthermore, the authors observed the anti-inflammatory effects of this intervention in peripheral lymph nodes [83].

Bifidobacteria comprise part of the human gut microbiota. In humans, it was observed that Bifidobacterium cell surface-associated exopolysaccharide has immunoregulatory effects, providing a diminished B-cell response. This effect was probably achieved by the protection from colonization with pathogens, inducing the upregulation of the immune response [84].

One of the major problems limiting the MMF application is severe gastrointestinal intolerance. It was experimentally proven that germ-free mice showed no MMF-associated diarrhea. The administration of antibiotics prevented it, and the species responsible for this side effect were Proteobacteria (predominantly Escherichia and Shigella) [40]. In humans, the administration of MMF led to microbiota changes resulting in the upregulation of lipopolysaccharide production. As the gastrointestinal tract is a colonized space and the constant administration of antibiotics would be impossible or even harmful, a possible means of intervention might be microbiota transfer from healthy donors.

The success of such a therapy was described in a child after heart transplantation with recurrent Clostridium difficile infection. The authors claim that the acute infection was anteceded by dysbiosis, mainly caused by Proteobacteria, and it could be corrected with appropriate fecal microbiota transfer [85]. The dysbiosis is caused both by immunosuppressive treatment and frequent antibiotic administration that adversely influence microbiota diversity and deplete protective germs.

The transfer of microbiota aims to restore Bacteroidetes and Firmicutes and to eliminate Proteobacteria.

Up to now, the strategy of microbiota transfer in the aspect of graft rejection is still under development and is one of the possible directions. Interventions altering microbiota, such as microbiota transfer, probiotics, or even antibiotics in the case of germs evidently promoting allograft rejection, could be another future direction.

## 9. Summary

The colonization of the gastrointestinal tract with bacteria is unequivocally connected with immunomodulation. Some bacteria seem to be immunomodulatory silent, others seem to upregulate innate and adaptive responses, and some downregulate the responses, probably by preventing colonization by pathogenic bacteria. Many immunological mechanisms are involved in these complex interactions. Despite distant localization, the processes taking place in the gastrointestinal tract and connected to the lymphatic tissue have an impact on remotely located transplanted organs. Animal models and observations in humans show that microbiota can influence graft survival and tolerance. Active strategies of immunomodulation with particular germs in humans are under development and seem to have some benefit in the future as a part of complex, up-to-date immunosuppressive treatments [85].

## Figures and Tables

**Table 1 biomedicines-11-01588-t001:** Current diagnostic methods for gut microbiota.

Diagnostics Methods	Advantages	Disadvantages
Samples from feces	Non-invasive, natural, inexpensive, suitable biomass for analysis, repeatable, convenient	Uneven distribution of bacteria in the feces may lead to erroneous results
Samples from endoscopy 1. Biopsy	Controlled sampling site, ability to obtain an accurate description of the tissue-associated microflora	Invasive, expensive, time consuming, and unpleasant for patients, contamination of the sample may occur, need to prepare the patient accordingly, insufficient biomass sample
Samples from endoscopy 2. Luminal brushing	Controlled sampling site, ability to obtain an accurate description of the tissue-associated microflora	Invasive (less invasive than a biopsy), expensive, time consuming, and unpleasant for patients, contamination of the sample may occur, need to prepare the patient accordingly
Samples from endoscopy 3. Laser capture microdissection	Controlled sampling site, ability to obtain an accurate description of the tissue-associated microflora and host–microbe interactions	Invasive, expensive, time consuming, and unpleasant for patients, contamination of the sample may occur, need to prepare the patient accordingly, insufficient biomass sample, is not appropriate for healthy control
Samples from aspirated intestinal fluid (catheter aspiration)	Controlled sampling site, ability to obtain an accurate description of the tissue-associated microflora	Invasive, time consuming, and unpleasant for patients, contamination of the sample may occur, need to prepare the patient accordingly
Samples from surgery	Controlled sampling site, ability to obtain an accurate description of the tissue-associated microflora, no contamination	Possible influence of preoperative preparations on the microflora, very invasive, is not appropriate for healthy control
Ingestible sampling devices (intelligent capsule)	Non-invasive, ability to obtain an accurate description of the tissue-associated microflora, no need to properly prepare the patient, sample contamination does not occur	Technically difficult and expensive
In vivo model (patients underwent ileostomy)	Non-invasive, inexpensive, sample contamination does not occur, repeatable and convenient sampling, sufficient biomass for analysis	Abnormal intestinal anatomy, is not appropriate for healthy control
Biology-related instruments (FISH)	Ability to obtain an accurate description of the tissue-associated microflora and host–microbe interactions	Not suited to a complex microbiome, the sampling device must be designed in advance

## Data Availability

All data stated in this review are available in the references cited.

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
