# Peer review of "Gut Microbiome in Patients after Heart Transplantation—Current State of Knowledge"

_biomedicines, 2023, doi:10.3390/biomedicines11061588_

Round 1

Reviewer 1 Report

Dear Editor,

I have read with great interest the manuscript entitled "Gut microbiome in patients after heart transplantation- current state of knowledge" proposed by Olek et al. However, there are some issues that need to be addressed before further processing. 

Introduction: some of the information is too generic. In the paragraph where the authors explain the basics of microbiota, the text is missing references. I suggest citing this recent paper that has very updated information for the general public on the topic - DOI: 10.1152/ajpgi.00161.2019). I also suggest introducing the concept of eu/dysbiosis before proposing how some factors related to HF may change gut microbiota composition. 

The remaining text mainly reports evidence of dysbiosis after heart transplantation. I suggest the author add a section on dysbiosis in heart-failure pre transplantation and maybe compare data before vs. after. 

Author Response

Thank you for your valuable comments. We modified the text according to your suggestions.

We introduced the term dysbiosis and added the suggested citation, as is important for the whole analysis.

We developed the subsection of dysbiosis in patients before heart transplantation. Indeed, it is better to present the changes chronologically and in the context of patients with heart failure.

Reviewer 2 Report

Olek et al described up-to-date microbiota diagnostic methods, interactions between microbiota and immunosuppressive drugs, immunomodulatory effects of dysbiosis and available strategies to modulate host's microbiota. Although the topic is intriguing, several concerns should be raised.

  1. There are many grammar mistakes; all the manuscripts should be carefully revised.
  2. There are many missing that should be revised.
  3. It would be better if the current diagnostic methods for gut microbiota were presented in a summarised figure.
  4. The pro- and anti-inflammatory effects of gut microbiota should be illustrated by a schematic diagram.

Author Response

Thank you for your suggestions. I hope, that will make the text more interesting and transparent to the reader.

The text had linguistic corrections by the native speaker. We added some significant topics and valuable literature citations.

We summarized diagnostic methods and pro- and anti-inflammatory effects in more readable form.

Reviewer 3 Report

The manuscript suffers from poor writing, structure, and lacks a specific conclusion. There are already many reviews on this topic that provide more informative and well-structured analyses. I believe that the manuscript would benefit from a more thorough literature review, improved organization, and a clearer focus on the research question.

The manuscript suffers from poor writing, structure, and lacks a specific conclusion.

Author Response

Thang you for your review. We hope that our corrections will improve the text.

We focused more on dysbiosis after heart transplantation. We added valuable literature positions and presented the text in more readable form.

Round 2

Reviewer 1 Report

The authors have answered the reviewers request and the paper is now ready for publication. 

Reviewer 3 Report

I am deeply disappointed to find that the authors have not taken my previous comments seriously and have failed to make substantial revisions to the manuscript. Despite my initial feedback highlighting the need for more rigorous and comprehensive analysis, the authors have only made superficial changes that do not address the fundamental shortcomings of the paper.

The manuscript still lacks a specific conclusion that can be drawn from the extensive collection of papers. It appears as though the authors have merely compiled a list of references without providing any significant insights or synthesizing the information in a meaningful way. The lack of a coherent and conclusive message diminishes the overall value and contribution of the manuscript to the field.

Furthermore, the authors have not made any effort to delve deeper into the subject matter or critically analyze the findings presented in the gathered literature. The paper remains at a surface level, merely presenting information without providing a comprehensive understanding or offering novel perspectives.

Therefore, I firmly maintain my decision to reject the manuscript, as the authors have not adequately addressed the concerns raised in my previous review. The lack of a specific conclusion, combined with the superficial nature of the analysis, significantly diminishes the scholarly merit of the paper.

The manuscript suffers from poor writing, and structure, and lacks a specific conclusion.